# Quantifying concordant genetic effects of de novo mutations on multiple disorders

**Hanmin Guo[1,2], Lin Hou[1,2,3], Yu Shi[4], Sheng Chih Jin[5], Xue Zeng[6,7], Boyang Li[8], Richard P Lifton[6,7], Martina Brueckner[6,9], Hongyu Zhao[6,8,10]\*, Qiongshi Lu[11]\***

[1]Center for Statistical Science, Tsinghua University, Beijing, China; [2]Department of Industrial Engineering, Tsinghua University, Beijing, China; [3]MOE Key Laboratory of Bioinformatics, School of Life Sciences, Tsinghua University, Beijing, China; [4]Yale School of Management, Yale University, New Haven, United States; [5]Department of Genetics, Washington University in St. Louis, St. Louis, United States; [6]Department of Genetics, Yale University, New Haven, United States; [7]Laboratory of Human Genetics and Genomics, Rockefeller University, New York, United States; [8]Department of Biostatistics, Yale School of Public Health, New Haven, United States; [9]Department of Pediatrics, Yale University, New Haven, United States; [10]Program of Computational Biology and Bioinformatics, Yale University, New Haven, United States; [11]Department of Biostatistics and Medical Informatics, University of Wisconsin-Madison, Madison, United States

**\*For correspondence:**
Hongyu.Zhao@yale.edu (HZ);
qlu@biostat.wisc.edu (QL)

**Competing interest:** The authors declare that no competing interests exist.

**Abstract** Exome sequencing on tens of thousands of parent-proband trios has identified numerous deleterious de novo mutations (DNMs) and implicated risk genes for many disorders. Recent studies have suggested shared genes and pathways are enriched for DNMs across multiple disorders. However, existing analytic strategies only focus on genes that reach statistical significance for multiple disorders and require large trio samples in each study. As a result, these methods are not able to characterize the full landscape of genetic sharing due to polygenicity and incomplete penetrance. In this work, we introduce EncoreDNM, a novel statistical framework to quantify shared genetic effects between two disorders characterized by concordant enrichment of DNMs in the exome. EncoreDNM makes use of exome-wide, summary-level DNM data, including genes that do not reach statistical significance in single-disorder analysis, to evaluate the overall and annotation-partitioned genetic sharing between two disorders. Applying EncoreDNM to DNM data of nine disorders, we identified abundant pairwise enrichment correlations, especially in genes intolerant to pathogenic mutations and genes highly expressed in fetal tissues. These results suggest that EncoreDNM improves current analytic approaches and may have broad applications in DNM studies.

## Editor's evaluation

Lu et al. provide a powerful statistical method that measures excess sharing of de novo mutations between pairs of disorders. This method extends the concept of 'genetic correlation' to disorders caused by de-novo mutations, measuring the correlation in excess de-novo mutations in genome-wide genes for different classes of mutations. The authors apply the method to nine disorders including a developmental disorder, autism spectrum disorder, congenital heart disease, schizophrenia, and intellectual disability, finding a statistically significant overlap between 12 pairs of disorders in de novo mutations that cause a loss of gene function. This method will be of interest to researchers working on disorders caused by de-novo mutations.

## Introduction

De novo mutations (DNMs) can be highly deleterious and provide important insights into the genetic cause for disease (*Veltman and Brunner, 2012*). As the cost of sequencing continues to drop, whole-exome sequencing (WES) studies conducted on tens of thousands of family trios have pinpointed numerous risk genes for a variety of disorders (*Lelieveld et al., 2016*; *Kaplanis et al., 2020*; *Satterstrom et al., 2020*). In addition, accumulating evidence suggests that risk genes enriched for pathogenic DNMs may be shared by multiple disorders (*Hoischen et al., 2014*; *Fromer et al., 2014*; *Homsy et al., 2015*; *Li et al., 2016*; *Nguyen et al., 2020*). These shared genes could reveal biological pathways that play prominent roles in disease etiology and shed light on clinically heterogeneous yet genetically related diseases (*Homsy et al., 2015*; *Li et al., 2016*; *Nguyen et al., 2020*).

Most efforts to identify shared risk genes directly compare genes that are significantly associated with each disorder (*Nguyen et al., 2017*; *Willsey et al., 2018*). There have been some successes with this approach in identifying shared genes and pathways (e.g. chromatin modifiers) underlying developmental disorder (DD), autism spectrum disorder (ASD), and congenital heart disease (CHD), thanks to the large trio samples in these studies (*Kaplanis et al., 2020*; *Satterstrom et al., 2020*; *Jin et al., 2017*), whereas findings in smaller studies remain suggestive (*Allen et al., 2013*; *Jin et al., 2020b*). Even in the largest studies to date, statistical power remains moderate for risk genes with weaker effects (*Kaplanis et al., 2020*; *Howrigan et al., 2020*). It is estimated that more than 1000 haploinsufficient genes contributing to developmental disorder risk have not yet achieved statistical significance in large WES studies (*Kaplanis et al., 2020*). Therefore, analytic approaches that only account for top significant genes cannot capture the full landscape of genetic sharing in multiple disorders. Recently, a Bayesian framework named mTADA was proposed to jointly analyze DNM data of two diseases and improve risk gene mapping (*Nguyen et al., 2020*). Although mTADA produces estimates for the proportion of shared risk genes, the statistical property of these parameter estimates has not been studied. There is a pressing need for powerful, robust, and interpretable methods that quantify concordant DNM association patterns for multiple disorders using exome-wide DNM counts.

Recent advances in estimating genetic correlations using summary data from genome-wide association studies (GWAS) may provide a blueprint for approaching this problem in DNM research (*Zhang et al., 2021a*). Modeling 'omnigenic' associations as independent random effects, linear mixed-effects models leverage genome-wide association profiles to quantify the correlation between additive genetic components of multiple complex traits (*Lee et al., 2012*; *Bulik-Sullivan et al., 2015*; *Lu et al., 2017*; *Ning et al., 2020*). These methods have identified ubiquitous genetic correlations across many human traits and revealed significant and robust genetic correlations that could not be inferred from significant GWAS associations alone (*Shi et al., 2017*; *Brainstorm, 2018*; *Guo et al., 2021*; *Zhang et al., 2021b*).

Here, we introduce EncoreDNM (**En**richment **cor**relation **e**stimator for **D**e **N**ovo **M**utations), a novel statistical framework that leverages exome-wide DNM counts, including genes that do not reach exome-wide statistical significance in single-disorder analysis, to estimate concordant DNM associations between disorders. EncoreDNM uses a generalized linear mixed-effects model to quantify the occurrence of DNMs while accounting for de novo mutability of each gene and technical inconsistencies between studies. We demonstrate the performance of EncoreDNM through extensive simulations and analyses of DNM data of nine disorders.

## Results

### Method overview

DNM counts in the exome deviate from the null (i.e. expected counts based on de novo mutability) when mutations play a role in disease etiology. Disease risk genes will show enrichment for deleterious DNMs in probands and non-risk genes may be slightly depleted for DNM counts. Our goal is to estimate the correlation of such deviation between two disorders, which we refer to as the DNM enrichment correlation. More specifically, we use a pair of mixed-effects Poisson regression models (*Munkin and Trivedi, 1999*) to quantify the occurrence of DNMs in two studies.

$$\begin{bmatrix} Y_{i1} \\ Y_{i2} \end{bmatrix} \sim \mathrm{Poisson}\left( \begin{bmatrix} \lambda_{i1} \\ \lambda_{i2} \end{bmatrix} \right),$$

$$\log\left( \begin{bmatrix} \lambda_{i1} \\ \lambda_{i2} \end{bmatrix} \right) = \begin{bmatrix} \beta_1 \\ \beta_2 \end{bmatrix} + \log\left( \begin{bmatrix} 2N_1 m_i \\ 2N_2 m_i \end{bmatrix} \right) + \begin{bmatrix} \phi_{i1} \\ \phi_{i2} \end{bmatrix},$$

$$\begin{bmatrix} \phi_{i1} \\ \phi_{i2} \end{bmatrix} \sim \mathrm{MVN}\left( \begin{bmatrix} 0 \\ 0 \end{bmatrix}, \begin{bmatrix} \sigma_1^2 & \rho\sigma_1\sigma_2 \\ \rho\sigma_1\sigma_2 & \sigma_2^2 \end{bmatrix} \right).$$

Here, $Y_{i1}, Y_{i2}$ are the DNM counts for the $i$-th gene and $N_1, N_2$ are the number of parent-proband trios in two studies, respectively. The log Poisson rates of DNM occurrence are decomposed into three components: the elevation component, the background component, and the deviation component. The elevation component $\beta_k$ ($k = 1, 2$) is a fixed effect term adjusting for systematic, exome-wide bias in DNM counts. One example of such bias is the batch effect caused by different sequencing and variant calling pipelines in two studies. The elevation parameter $\beta_k$ tends to be positive when DNMs are over-called with higher sensitivity and negative when DNMs are under-called with higher specificity (**Wei et al., 2015**). The background component $\log\left(2N_k m_i\right)$ is a gene-specific fixed effect that reflects the expected mutation counts determined by the genomic sequence context under the null (**Samocha et al., 2014**). $m_i$ is the de novo mutability for the $i$-th gene, and $2N_1 m_i$ and $2N_2 m_i$ are the expected DNM counts in the $i$-th gene under the null in two studies. The deviation component $\phi_{ik}$ is a gene-specific random effect that quantifies the deviation of DNM profile from what is expected under the null (i.e. no risk genes for the disorder). $\phi_{i1}$ and $\phi_{i2}$ follow a multivariate normal distribution with dispersion parameters $\sigma_1$ and $\sigma_2$ and a correlation $\rho$. A larger value of the dispersion parameter $\sigma_k$ indicates a more substantial deviation from the null. That is, DNM counts show strong enrichment in some genes and depletion in other genes compared to the expectation based on de novo mutability. A smaller value of $\sigma_k$ suggests that the DNM count data is well in line with what is expected based on

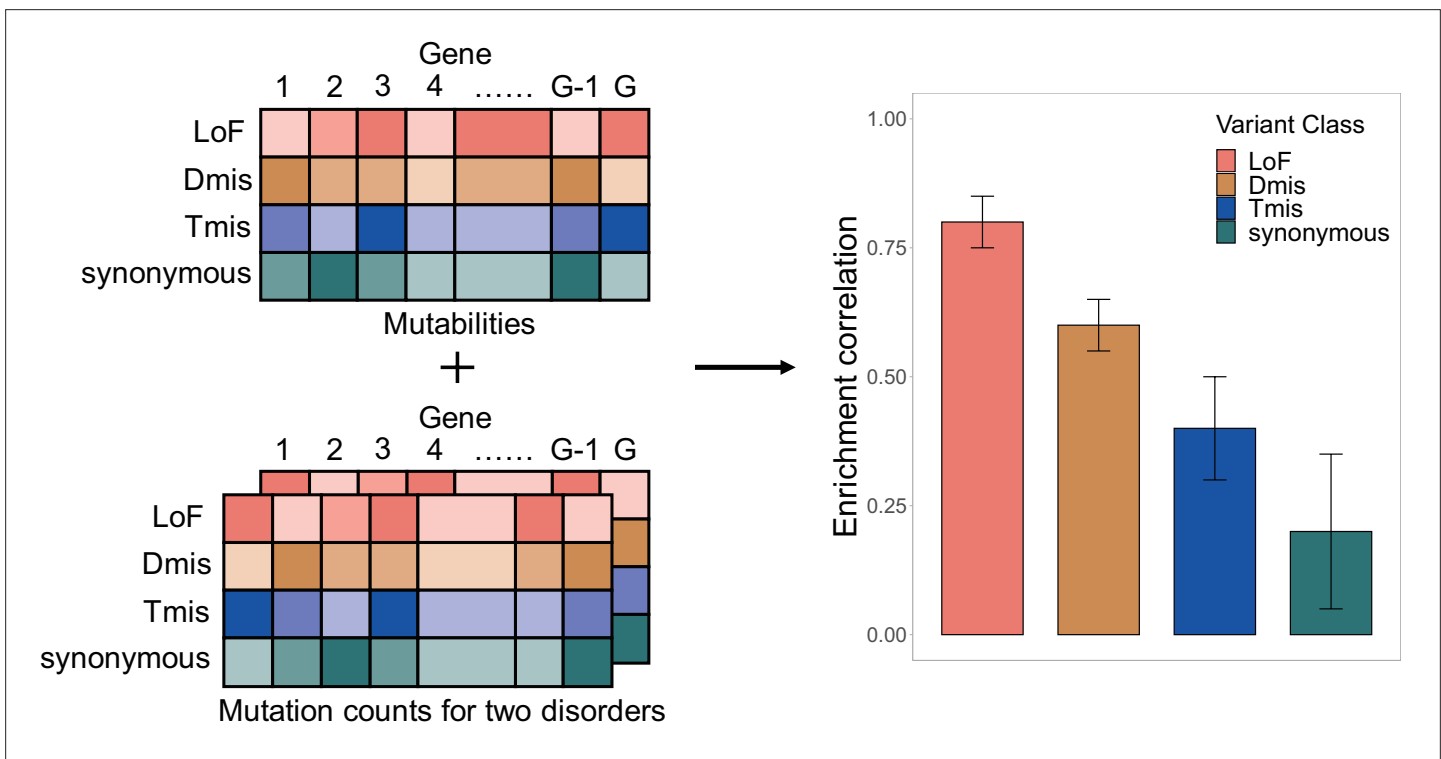

**Figure 1.** EncoreDNM workflow. The inputs of EncoreDNM are de novo mutability of each gene and exome-wide, annotated DNM counts from two studies. We fit a mixed-effects Poisson model to estimate the DNM enrichment correlation between two disorders for each variant class.

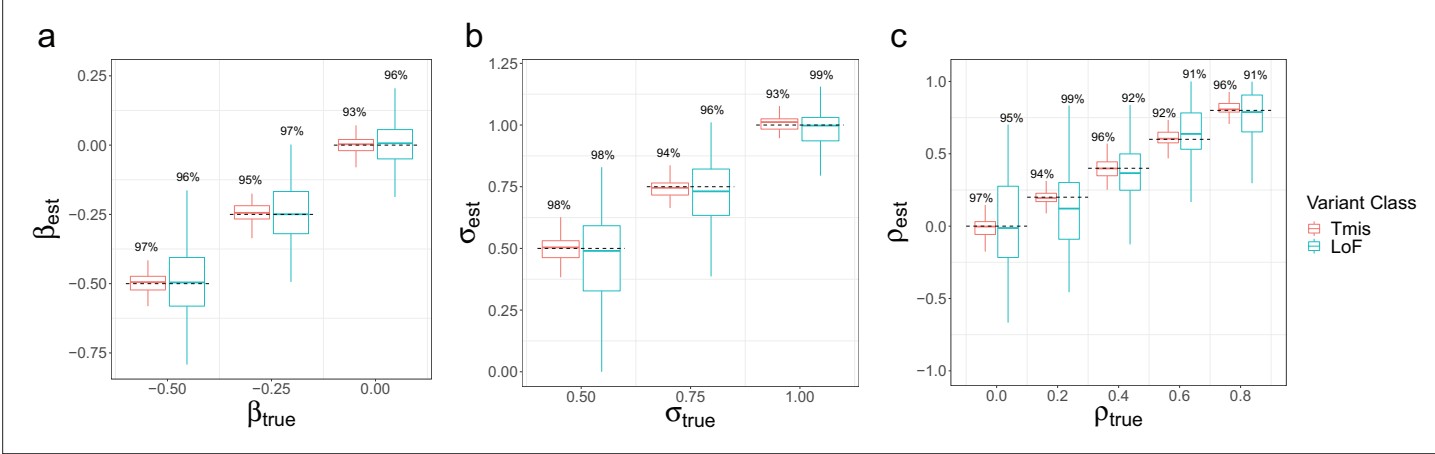

**Figure 2.** Parameter estimation results of EncoreDNM. (**a**) Boxplot of $\beta$ estimates in single-trait analysis with $\sigma$ fixed at 0.75. (**b**) Boxplot of $\sigma$ estimates in single-trait analysis with $\beta$ fixed at –0.25. (**c**) Boxplot of $\rho$ estimates in cross-trait analysis with $\beta$ and $\sigma$ fixed at –0.25 and 0.75. True parameter values are marked by dashed lines. The number above each box represents the coverage rate of 95% Wald confidence intervals. Each simulation setting was repeated 100 times.

The online version of this article includes the following figure supplement(s) for figure 2:

**Figure supplement 1.** Estimation results of elevation parameter $\beta$ under a mixed-effects Poisson regression model.

**Figure supplement 2.** Estimation results of dispersion parameter $\sigma$ under a mixed-effects Poisson regression model.

the null model. DNM enrichment correlation is denoted by $\rho$ and is our main parameter of interest. It quantifies the concordance of DNM burden in two disorders.

Parameters in this model can be estimated using a Monte Carlo maximum likelihood estimation (MLE) procedure. Standard errors of the estimates are obtained through inversion of the observed Fisher information matrix. In practice, we use annotated DNM data as input and fit mixed-effects Poisson models for each variant class separately: loss of function (LoF), deleterious missense (Dmis, defined as MetaSVM-deleterious), tolerable missense (Tmis, defined as MetaSVM-tolerable), and synonymous (*Figure 1*). More details about model setup and parameter estimation are discussed in Materials and methods.

## Simulation results

We conducted simulations to assess the parameter estimation performance of EncoreDNM in various settings. We focused on two variant classes, that is, Tmis and LoF variants, since they have the highest and lowest median mutabilities in the exome. We used EncoreDNM to estimate the elevation parameter $\beta$, dispersion parameter $\sigma$, and enrichment correlation $\rho$ (Materials and methods). Under various parameter settings, EncoreDNM always provided unbiased estimation of the parameters (*Figure 2* and *Figure 2—figure supplements 1–2*). Furthermore, the 95% Wald confidence intervals achieved coverage rates close to 95% under all simulation settings, demonstrating the effectiveness of EncoreDNM to provide accurate statistical inference.

Next, we compared the performance of EncoreDNM with mTADA (*Nguyen et al., 2020*), a Bayesian framework that estimates the proportion of shared risk genes for two disorders. First, we simulated DNM data under the mixed-effects Poisson model. We evaluated two methods across a range of combinations of elevation parameter, dispersion parameter, and sample size for two disorders. The false positive rates for our method were well-calibrated in all parameter settings, but mTADA produced false positive findings when the observed DNM counts were relatively small (e.g. due to reduced elevation or dispersion parameters or a lower sample size; *Figure 3a*). We also assessed the statistical power of two approaches under a baseline setting where false positives for both methods were controlled. As enrichment correlation increased, EncoreDNM achieved universally greater statistical power compared to mTADA (*Figure 3b*).

To ensure a fair comparison, we also considered a mis-specified model setting where we randomly distributed the total DNM counts for each disorder into all genes with an enrichment in causal genes (Materials and methods). EncoreDNM showed well-controlled false positive rate across all simulation

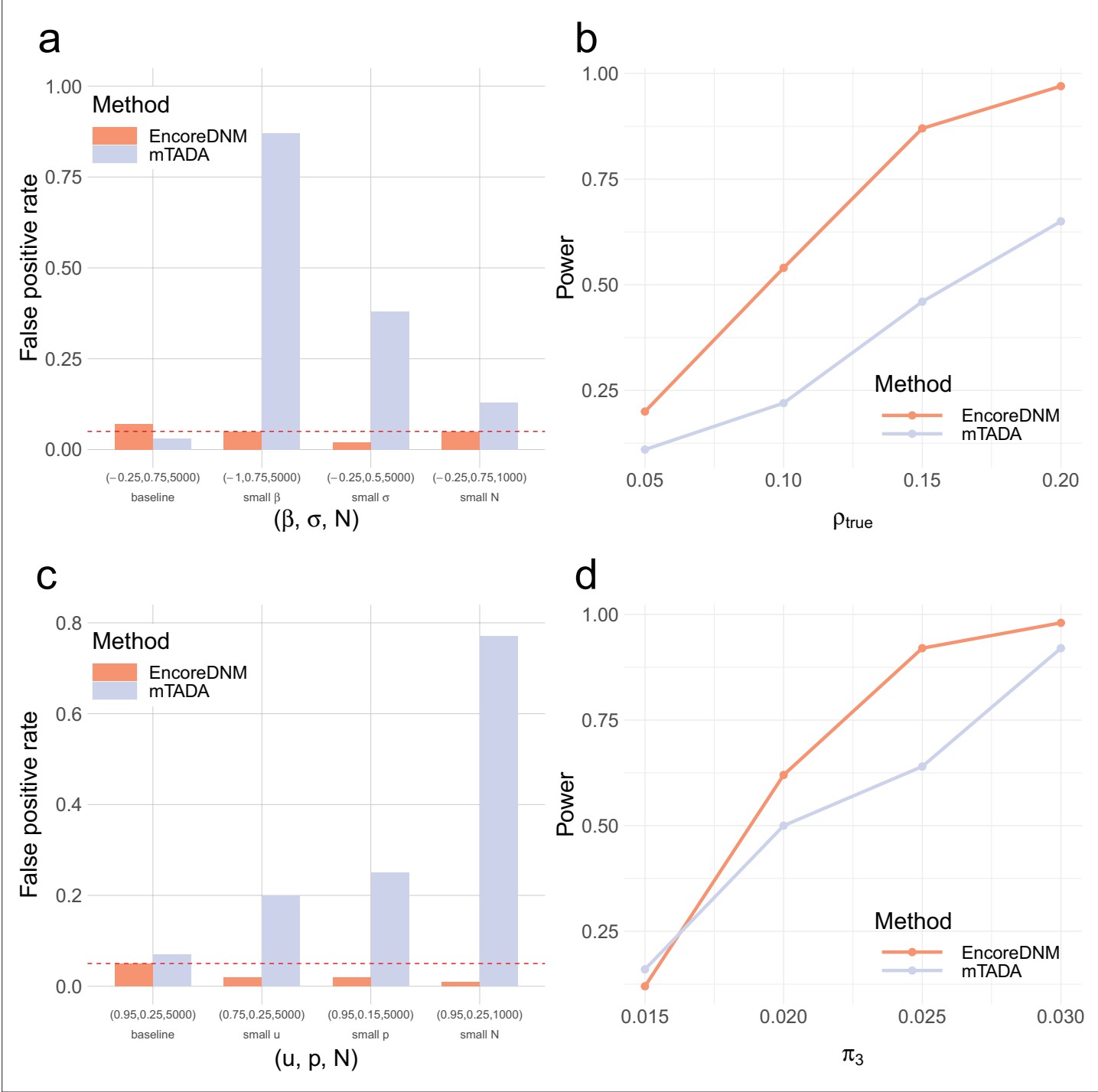

**Figure 3.** Comparison of EncoreDNM and mTADA. (**a**) False positive rates under a mixed-effects Poisson regression model. (**b**) Statistical power of two methods under a mixed-effects Poisson regression model as the enrichment correlation increases. Parameters $(\beta, \sigma, N)$ were fixed at (–0.25, 0.75, 5000) for both disorders. (**c**) False positive rates under a multinomial model. (**d**) Statistical power under a multinomial model with varying proportion of shared causal genes. Parameters $(u, p, N)$ were fixed at (0.95, 0.25, 5000) for both disorders. Each simulation setting was repeated 100 times.

settings, whereas severe inflation of false positives arose for mTADA when the total mutation count, the proportion of probands that can be explained by DNMs, or the sample size were small (**Figure 3c**). Furthermore, we compared the statistical power of two methods under this model in a baseline setting where false positive rate was controlled. EncoreDNM showed higher statistical power compared to mTADA as the fraction of shared causal genes increased (**Figure 3d**).

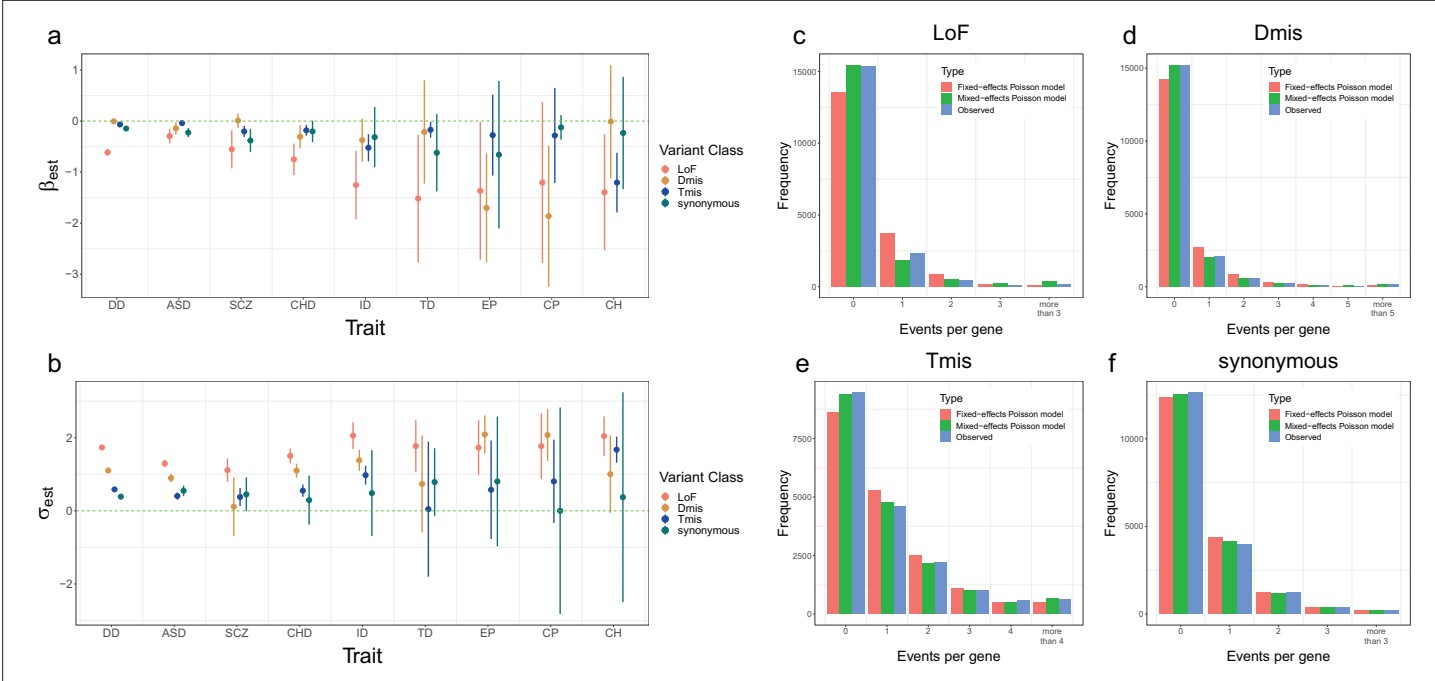

**Figure 4.** Model fitting results for nine disorders. (**a, b**) Estimation results of $\beta$ and $\sigma$ for nine disorders and four variant classes. Error bars represent 1.96*standard errors. Sample sizes of DNM datasets for each disorder are provided in *Supplementary file 1*-STable 1. (**c–f**) Distribution of DNM events per gene in four variant classes for developmental disorder. Red and green bars represent the expected frequency of genes under the fixed-effects and mixed-effects Poisson regression models, respectively. Blue bars represent the observed frequency of genes.

The online version of this article includes the following figure supplement(s) for figure 4:

**Figure supplement 1.** Likelihood ratio test shows significantly improved goodness of fit of the mixed-effects Poisson model compared to a fixed-effects model without the deviation component.

## Pervasive enrichment correlation of damaging DNMs among developmental disorders

We applied EncoreDNM to DNM data of nine disorders (*Supplementary file 1*-STable 1; Materials and methods): developmental disorder (n=31,058; number of trios; *Kaplanis et al., 2020*), autism spectrum disorder (n=6430; *Satterstrom et al., 2020*), schizophrenia (SCZ; n=2772; *Howrigan et al., 2020*), congenital heart disease (n=2645; *Jin et al., 2017*), intellectual disability (ID; n=820; *Lelieveld et al., 2016*), Tourette disorder (TD; n=484) *Willsey et al., 2017*, epileptic encephalopathies (EP; n=264; *Allen et al., 2013*), cerebral palsy (CP; n=250; *Jin et al., 2020b*), and congenital hydrocephalus (CH; n=232; *Jin et al., 2020a*). In addition, we also included 1789 trios comprising healthy parents and unaffected siblings of autism probands as controls (*Krumm et al., 2015*).

We first performed single-trait analysis under the mixed-effects Poisson model for each disorder. The estimated elevation parameters (i.e. $\beta$) were negative for almost all disorders and variant classes (*Figure 4a*), with LoF variants showing particularly lower parameter estimates. This may be explained by more stringent quality control in LoF variant calling (*Jin et al., 2017*) and potential survival bias (*Lek et al., 2016*). It is also consistent with a depletion of LoF DNMs in healthy control trios (*Homsy et al., 2015*). The dispersion parameter estimates (i.e. $\sigma$) were higher for LoF variants than other variant classes (*Figure 4b*), which is consistent with our expectation that LoF variants have stronger effects on disease risk and should show a larger deviation from the null mutation rate in disease probands. We also compared the goodness of fit of our proposed mixed-effects Poisson model to a simpler fixed-effects model without the deviation component (Materials and methods). The expected distribution of recurrent DNM counts showed substantial and statistically significant improvement under the mixed-effects Poisson model (*Figure 4c–f*, *Figure 4—figure supplement 1*, and *Supplementary file 1*-STable 2).

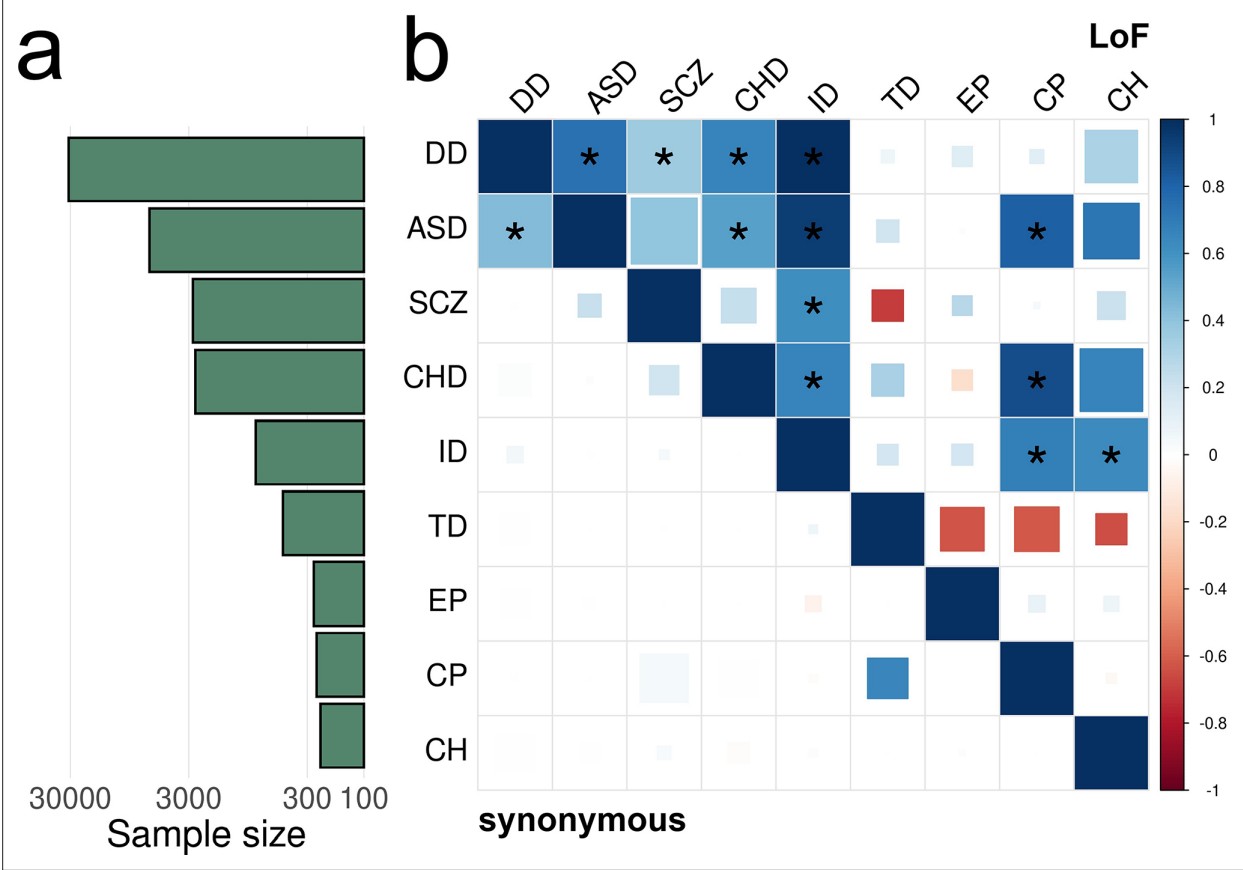

**Figure 5.** EncoreDNM identifies pervasive enrichment correlations of damaging DNMs among nine disorders. (**a**) Shows sample size (for example, number of trios) for each disease. X-axis denotes sample size on the log scale. (**b**) Heatmap of enrichment correlations for LoF (upper triangle) and synonymous (lower triangle) DNMs among nine disorders. Larger squares represent more significant p-values, and deeper color represents stronger correlations. Significant correlations (FDR <0.05) are shown as full-sized squares marked by asterisks.

The online version of this article includes the following figure supplement(s) for figure 5:

**Figure supplement 1.** DNM enrichment correlations of nine disorders based on Dmis and Tmis variants.

**Figure supplement 2.** DNM enrichment correlations between nine disorders and controls.

**Figure supplement 3.** Number of significant correlations identified for each disorder is proportional to its sample size.

**Figure supplement 4.** Lollipop plot for LoF DNMs in *CTNNB1*.

**Figure supplement 5.** Lollipop plot for LoF DNMs in *FBXO11*.

**Figure supplement 6.** DNM genetic sharing in nine disorders estimated for LoF, Dmis, Tmis, and synonymous DNMs using mTADA.

**Figure supplement 7.** DNM genetic sharing in nine disorders and controls identified by mTADA.

**Figure supplement 8.** Comparison of GWAS- and DNM-based estimation of genetic sharing among five disorders.

**Figure supplement 9.** Group-wise jackknife method and inversion of Fisher information matrix method produced similar standard error estimates for LoF variants.

Next, we estimated pairwise DNM enrichment correlations for 9 disorders. In total, we identified 25 pairs of disorders with significant correlations at a false discovery rate (FDR) cutoff of 0.05 (*Figure 5* and *Figure 5—figure supplement 1*), including 12 significant correlations for LoF variants, 7 for Dmis variants, 5 for Tmis variants, and only 1 significant correlation for synonymous variants. Notably, all significant correlations are positive (*Supplementary file 1*-STable 3). No significant correlation was identified between any disorder and healthy controls (*Figure 5—figure supplement 2*). This is consistent with our expectation, since DNMs in the control groups will distribute proportionally according to the de novo mutability without showing enrichment in certain genes. The number of identified significant correlations for each disorder was proportional to the sample size in each study (Spearman correlation = 0.70) with controls being a notable outlier (*Figure 5—figure supplement 3*).

We identified highly concordant and significant LoF DNM enrichment among developmental disorder, autism, intellectual disability, and congenital heart disease, which is consistent with previous reports (*Li et al., 2016*; *Nguyen et al., 2020*; *Nguyen et al., 2017*; *Hormozdiari et al., 2015*). Schizophrenia shows highly significant LoF correlations with developmental disorder (p=2.0e-3) and intellectual disability (3.7e-5). The positive enrichment correlation between autism and cerebral palsy in LoF variants ($\rho$=0.81, p=3.3e-3) is consistent with their co-occurrence (*Christensen et al., 2014*). The high enrichment correlation between intellectual disability and cerebral palsy in LoF variants ($\rho$=0.68, p=1.0e-4) is consistent with the associations between intellectual disability and motor or non-motor abnormalities caused by cerebral palsy (*Reid et al., 2018*). A previous study also suggested significant genetic sharing of intellectual disability and cerebral palsy by overlapping genes harboring rare damaging variants (*Jin et al., 2020b*). Here, we obtained consistent results after accounting for de novo mutabilities and potential confounding bias.

Some significant correlations identified in our analysis are consistent with phenotypic associations in epidemiological studies, but have not been reported using genetic data to the extent of our knowledge. For example, the LoF enrichment correlation between congenital heart disease and cerebral palsy ($\rho$=0.88, p=1.7e-3) is consistent with findings that reduced supply of oxygenated blood in fetal brain due to cardiac malformations may be a risk factor for cerebral palsy (*Garne et al., 2008*). The enrichment correlation between intellectual disability and congenital hydrocephalus in LoF variants ($\rho$=0.63, p=2.4e-3) is consistent with lower intellectual performance in a proportion of children with congenital hydrocephalus (*Lumenta and Skotarczak, 1995*).

Genes showing pathogenic DNMs in multiple disorders may shed light on the mechanisms underlying enrichment correlations (*Supplementary file 1*-STable 4). We identified five genes (*CTNNB1*, *NBEA*, *POGZ*, *SPRED2*, and *KMT2C*) with LoF DNMs in five different disorders and 21 genes had LoF DNMs in four disorders (*Supplementary file 1*-STable 5). These 26 genes with LoF variants in at least four disorders were significantly enriched for 63 gene ontology (GO) terms with FDR <0.05 (*Supplementary file 1*-STable 6). Chromatin organization (p=7.8e-11), nucleoplasm (p=2.8e-10), chromosome organization (p=6.8e-10), histone methyltransferase complex (p=1.4e-9), and positive regulation of gene expression (p=2.2e-9) were the most significantly enriched GO terms. One notable example consistently included in these gene sets is *CTNNB1* (*Figure 5—figure supplement 4*). It encodes $\beta$-catenin, is one of the only two genes reaching genome-wide significance in a recent WES study for cerebral palsy (*Jin et al., 2020b*), and also harbors multiple LoF variants in developmental disorder, intellectual disability, autism, and congenital heart disease. It is a fundamental component of the canonical Wnt signaling pathway which is known to confer genetic risk for autism (*O'Roak et al., 2012*). Genes with recurrent damaging DNMs in multiple disorders also revealed shared biological function across these disorders (*Rees et al., 2021*). We identified 30 recurrent cross-disorder LoF mutations that were not recurrent in developmental disorder alone (*Supplementary file 1*). *FBXO11*, encoding the F-box only protein 31, shows two recurrent p.Ser831fs LoF variants in autism and congenital hydrocephalus (*Figure 5—figure supplement 5*; p=1.9e-3; Materials and methods). The F-box protein constitutes a substrate-recognition component of the SCF (SKP1-cullin-F-box) complex, an E3-ubiquitin ligase complex responsible for ubiquitination and proteasomal degradation (*Cardozo and Pagano, 2004*). DNMs in *FBXO11* have been previously implicated in severe intellectual disability individuals with autistic behavior problem (*Jansen et al., 2019*) and neurodevelopmental disorder (*Gregor et al., 2018*).

For comparison, we also applied mTADA to the same nine disorders and control trios. In total, mTADA identified 117 disorder pairs with significant genetic sharings at an FDR cutoff of 0.05 (*Supplementary file 1*-STable 8 and *Figure 5—figure supplement 6*). Notably, we identified significant synonymous DNM correlations for all 36 disorder pairs and between all disorders and healthy controls (*Figure 5—figure supplement 7*). These results are consistent with the simulation results and suggest a substantially inflated false positive rate in mTADA.

## Partitioning DNM enrichment correlation by gene set

To gain biological insights into the shared genetic architecture of nine disorders, we repeated EncoreDNM correlation analysis in several gene sets. First, we defined genes with high/low probability of intolerance to LoF variants using pLI scores (*Karczewski et al., 2020*), and identified genes with high/low brain expression (HBE/LBE) (*Werling et al., 2020*; Materials and methods; *Supplementary*

file 1-STable 9). We identified 11 and 12 disorder pairs showing significant enrichment correlations for LoF DNMs in high-pLI genes and HBE genes, respectively (*Figure 6a–b*). We observed fewer significant correlations for Dmis and Tmis variants in these gene sets (*Figure 6—figure supplements 1–2*). All identified significant correlations were positive (*Supplementary file 1*-STables 10 -11). No significant correlations were identified for synonymous variants (*Figure 6—figure supplements 1–2*) or between disorders and controls (*Figure 6—figure supplements 3–4*).

We observed a clear enrichment of significant correlations in disease-relevant gene sets. Overall, high-pLI genes showed substantially stronger correlations across disorders than genes with low pLI (one-sided Kolmogorov-Smirnov test; p=2.3e-6). Similarly, enrichment correlations were stronger in HBE genes than in LBE genes (p=8.8e-7). Among the 11 disorder pairs showing significant enrichment correlations in high-pLI genes, two pairs, that is, autism-schizophrenia ($\rho$=0.68, p=2.4e-3) and developmental disorder-congenital hydrocephalus ($\rho$=0.43, p=1.5e-3), were not identified in the exome-wide analysis. We also identified four novel disorder pairs with significant correlations in HBE genes, including developmental disorder-cerebral palsy ($\rho$=0.80, p=9.5e-5), developmental disorder-congenital hydrocephalus ($\rho$=0.67, p=1.4e-3), autism-congenital hydrocephalus ($\rho$=0.82, p=4.7e-4), and schizophrenia-epileptic encephalopathies ($\rho$=0.66, p=2.0e-3). These novel enrichment correlations are consistent with known comorbidities between these disorders (*Kielinen et al., 2004*; *Kilincaslan and Mukaddes, 2009*) and findings based on significant risk genes (*Li et al., 2016*; *Jin et al., 2020a*; *Kume et al., 1998*; *Cao and Wu, 2015*).

Furthermore, we estimated DNM enrichment correlations in genes with high/low expression in mouse developing heart (HHE/LHE) (*Homsy et al., 2015*; Materials and methods; *Supplementary file 1*-STable 9). We identified 9 significant enrichment correlations for LoF variants in HHE genes (*Figure 6c*). Strength of enrichment correlations did not show a significant difference between HHE and LHE genes (p=0.846), possibly due to a lack of cardiac disorders in our analysis. Finally, we estimated enrichment correlations between congenital heart disease and other disorders in known pathways for congenital heart disease (*Zaidi and Brueckner, 2017*; Materials and methods; *Supplementary file 1*-STable 9). We identified five significant correlations for LoF variants (*Figure 6d*), including a novel correlation between congenital heart disease and Tourette disorder ($\rho$=0.93, p=3.3e-9). Of note, arrhythmia caused by congenital heart disease is a known risk factor for Tourette disorder (*Gulisano et al., 2011*). In these analyses, all significant enrichment correlations were positive (*Supplementary file 1*) and other variant classes showed generally weaker correlations than LoF variants (*Figure 6—figure supplements 5–6*). We did not observe significant correlations in these gene sets between disorders and controls (*Figure 6—figure supplements 7–8*).

## Discussion

In this paper, we introduced EncoreDNM, a novel statistical framework to quantify correlated DNM enrichment between two disorders. Through extensive simulations and analyses of DNM data for nine disorders, we demonstrated that our proposed mixed-effects Poisson regression model provides unbiased parameter estimates, shows well-controlled false positive rate, and is robust to exome-wide technical biases. Leveraging exome-wide DNM counts and genomic context-based mutability data, EncoreDNM achieves superior fit for real DNM datasets compared to simpler models and provides statistically powerful and computationally efficient estimation of DNM enrichment correlation. Further, EncoreDNM can quantify concordant genetic effects for user-defined variant classes within pre-specified gene sets, thus is suitable for exploring diverse types of hypotheses and can provide crucial biological insights into the shared genetic etiology in multiple disorders. In comparison, the Bayesian approach implemented in mTADA can produce false positives findings, especially when the DNM count is low, possibility due to the overestimated proportion of risk genes. We still observed inflation in false positive rates under a more stringent significance cutoff or using posterior probability threshold strategy (*Supplementary file 1*-STables 14-17).

Multi-trait analyses of GWAS data have revealed shared genetic architecture among many neuropsychiatric traits (*Brainstorm, 2018*; *Lee et al., 2013*; *Gratten et al., 2014*; *Abdellaoui and Verweij, 2021*). These findings have led to the identification of pleiotropic variants, genes, and hub genomic regions underlying many traits and have revealed multiple psychopathological factors jointly affecting human neurological phenotypes (*Lee, 2019*; *Wang et al., 2015*). Although emerging evidence suggests that causal DNMs underlying several disorders with well-powered studies (e.g. congenital

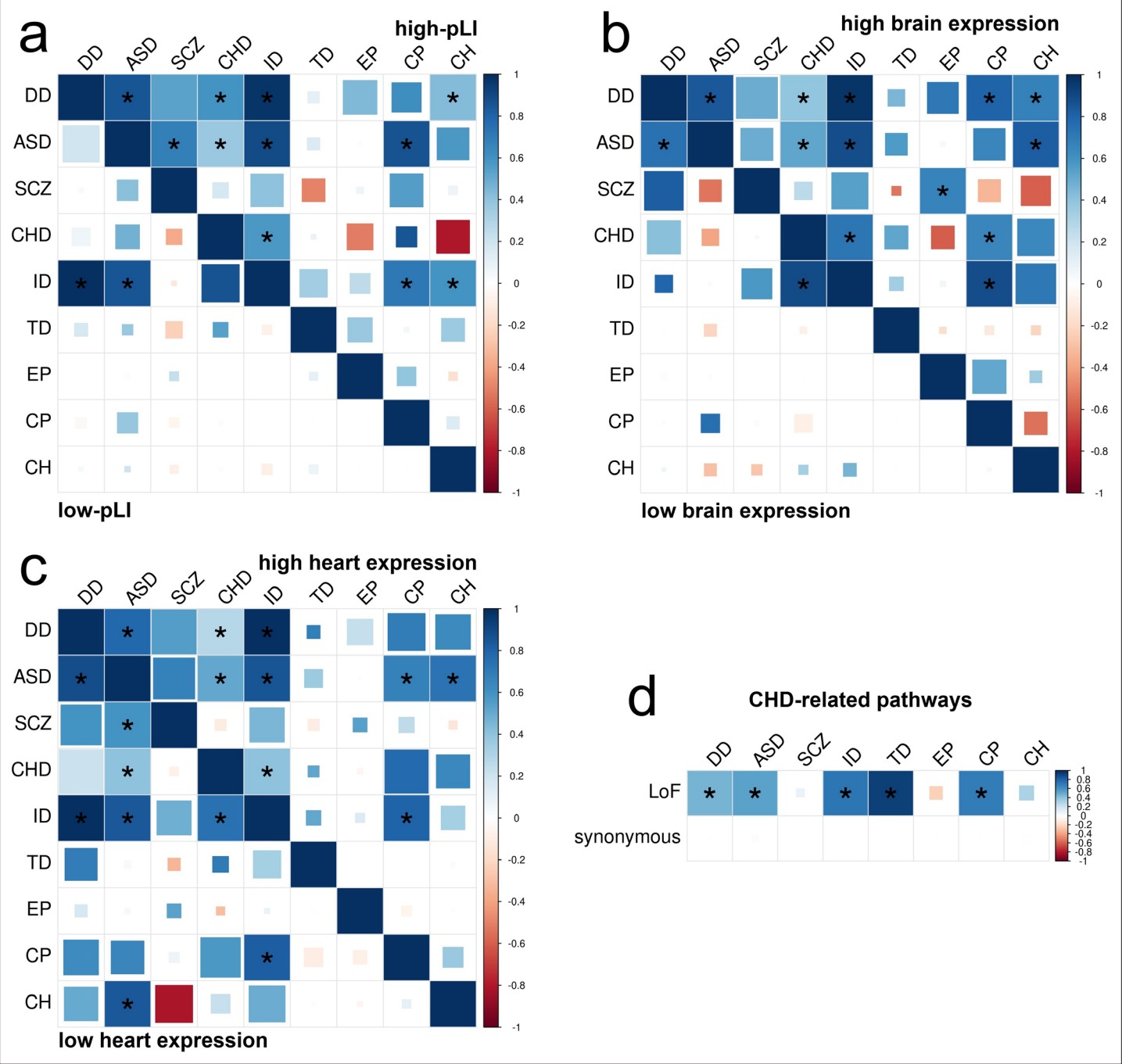

**Figure 6.** DNM enrichment correlations in disease-relevant gene sets. (**a**) Enrichment correlations in high-pLI genes (upper triangle) and low-pLI genes (lower triangle) for LoF variants. Here, pLI is the probability of being loss-of-function intolerant (see Materials and methods). (**b**) Enrichment correlations in HBE genes (upper triangle) and LBE genes (lower triangle) for LoF variants. (**c**) Enrichment correlations in HHE genes (upper triangle) and LHE genes (lower triangle) for LoF variants. (**d**) Enrichment correlations in CHD-related pathways for LoF and synonymous variants. Larger squares represent more significant p-values, and deeper color represents stronger correlations. Significant correlations (FDR <0.05) are shown as full-sized squares marked by asterisks.

The online version of this article includes the following figure supplement(s) for figure 6:

**Figure supplement 1.** DNM enrichment correlations in high-pLI genes (upper triangle) and low-pLI genes (lower triangle) for Dmis, Tmis, and synonymous variants.

**Figure supplement 2.** DNM enrichment correlations in HBE genes (upper triangle) and LBE genes (lower triangle) for Dmis, Tmis, and synonymous variants.

*Figure 6 continued on next page*

*Figure 6 continued*

**Figure supplement 3.** DNM enrichment correlations between nine disorders and controls in high-pLI and low-pLI gene sets.

**Figure supplement 4.** DNM enrichment correlations between nine disorders and controls in HBE and LBE genes.

**Figure supplement 5.** DNM enrichment correlations in HHE genes (upper triangle) and LHE genes (lower triangle) for Dmis, Tmis, and synonymous variants.

**Figure supplement 6.** DNM enrichment correlations in CHD-related pathways for Dmis and Tmis variants.

**Figure supplement 7.** DNM enrichment correlations between nine disorders and controls in HHE and LHE gene sets.

**Figure supplement 8.** DNM enrichment correlations between CHD and controls in CHD-related pathways.

heart disease and neurodevelopmental disorders; *Homsy et al., 2015*) may be shared, our understanding of the extent and the mechanism underlying such sharing remains incomplete. Applied to DNM data for nine disorders, EncoreDNM identified pervasive enrichment correlations of DNMs. We observed particularly strong correlations in pathogenic variant classes (e.g. LoF and Dmis variants) and disease-relevant genes (e.g. genes with high pLI and genes highly expressed in relevant tissues). Genes underlying these correlations were significantly enriched in pathways involved in chromatin organization and modification and gene expression regulation. The DNM correlations were substantially attenuated in genes with lower expression and genes with frequent occurrences of LoF variants in the population. A similar attenuation was observed in less pathogenic variant classes (e.g., synonymous variants). Further, no significant correlations were identified between any disorder and healthy controls. We also compared DNM enrichment correlations of five disorders with genetic correlations estimated from GWAS summary statistics (*Supplementary file 1*-STable 18). We had consistent findings from GWAS and DNM data (Spearman correlation = 0.70; *Figure 5—figure supplement 8* and *Supplementary file 1*-STable 19). These results lay the groundwork for future investigations of pleiotropic mechanisms of DNMs.

Our study has some limitations. First, EncoreDNM assumes probands from different input studies to be independent. In rare cases when two studies have overlapping proband samples, enrichment correlation estimates may be inflated and must be interpreted with caution. Second, genetic correlation methods based on GWAS summary data provided key motivations for the mixed-effects Poisson regression model in our study. Built upon genetic correlations, a plethora of methods have been developed in the GWAS literature to jointly model more than two GWAS (*Turley et al., 2018*), identify and quantify common factors underlying multiple traits (*Grotzinger et al., 2019*; *Grotzinger et al., 2020*), estimate causal effects among different traits (*Pickrell et al., 2016*), and identify pleiotropic genomic regions through hypothesis-free scans (*Guo et al., 2021*). Future directions of EncoreDNM include using enrichment correlation to improve gene discovery, learning the directional effects and the causal structure underlying multiple disorders, and dynamically searching for gene sets and annotation classes with shared genetic effects without pre-specifying the hypothesis.

Taken together, we provide a new analytic approach to an important problem in DNM studies. We believe EncoreDNM improves the statistical rigor in multi-disorder DNM modeling and opens up many interesting future directions in both method development and follow-up analyses in WES studies. As trio sample size in WES studies continues to grow, EncoreDNM will have broad applications and can greatly benefit DNM research.

## Materials and methods
### Statistical model
For a single study, we assume that DNM counts in a given variant class (for example, synonymous variants) follow a mixed-effects Poisson model:

$$Y_i \sim \text{Poisson}\left(\lambda_i\right),$$

$$\log\left(\lambda_i\right) = \beta + \log\left(2Nm_i\right) + \phi_i,$$

$$\phi_i \sim \text{N}\left(0, \sigma^2\right), \quad \text{for } i = 1, \dots, G,$$

where $Y_i$ is the DNM count in the $i$-th gene, $N$ is the number of trios, $m_i$ is the de novo mutability for the $i$-th gene (for example, mutation rate per chromosome per generation) which is known a priori (**Samocha et al., 2014**), and $G$ is the total number of genes in the study. The elevation parameter $\beta$ quantifies the global elevation of mutation rate compared to mutability estimates based on genomic sequence alone. Gene-specific deviation from expected DNM rate is quantified by random effect $\phi_i$ with a dispersion parameter $\sigma$. Here, the $\phi_i$ are assumed to be independent across different genes, in which case the observed DNM counts of different genes are independent. There is no constraint on the value of $\beta$, and the dispersion parameter $\sigma$ can be any positive value.

Next, we describe how we expand this model to quantify the shared genetics of two disorders. We adopt a flexible Poisson-lognormal mixture framework that can accommodate both overdispersion and correlation (**Munkin and Trivedi, 1999**). We assume DNM counts in a given variant class for two diseases follow:

$$\begin{bmatrix} Y_{i1} \\ Y_{i2} \end{bmatrix} \sim \text{Poisson}\left( \begin{bmatrix} \lambda_{i1} \\ \lambda_{i2} \end{bmatrix} \right),$$

$$\log\left( \begin{bmatrix} \lambda_{i1} \\ \lambda_{i2} \end{bmatrix} \right) = \begin{bmatrix} \beta_1 \\ \beta_2 \end{bmatrix} + \log\left( \begin{bmatrix} 2N_1 m_i \\ 2N_2 m_i \end{bmatrix} \right) + \begin{bmatrix} \phi_{i1} \\ \phi_{i2} \end{bmatrix},$$

$$\begin{bmatrix} \phi_{i1} \\ \phi_{i2} \end{bmatrix} \sim \text{MVN}\left( \begin{bmatrix} 0 \\ 0 \end{bmatrix}, \begin{bmatrix} \sigma_1^2 & \rho\sigma_1\sigma_2 \\ \rho\sigma_1\sigma_2 & \sigma_2^2 \end{bmatrix} \right),$$

where $Y_{i1}, Y_{i2}$ are the DNM counts for the $i$-th gene and $N_1, N_2$ are the trio sizes in two studies, respectively. Similar to the single-trait model, $m_i$ is the mutability for the $i$-th gene. $\beta_1, \beta_2$ are the elevation parameters, and $\phi_{i1}, \phi_{i2}$ are the gene-specific random effects with dispersion parameters $\sigma_1, \sigma_2$, for two disorders respectively. $\rho$ is the enrichment correlation which quantifies the concordance of the gene-specific DNM burden between two disorders. Here, $\beta_1, \beta_2, \sigma_1, \sigma_2, \rho$ are unknown parameters. The gene specific effects for two disorders are assumed to be independent for different genes. We also assume that there is no shared sample for two disorders, in which case $Y_{i1}$ is independent with $Y_{i2}$ given $\begin{bmatrix} \lambda_{i1} \\ \lambda_{i2} \end{bmatrix}$.

## Parameter estimation

We implement an MLE procedure to estimate unknown parameters. For single-trait analysis, the log-likelihood function can be expressed as follows:

$$l\left(\beta, \sigma | \boldsymbol{Y}\right) = \sum_{i=1}^{G} \log\left[ \int \exp\left(-\lambda_i\right) \lambda_i^{Y_i} * f\left(\phi_i\right) d\phi_i \right] + C,$$

where $\boldsymbol{Y} = \begin{bmatrix} Y_1, \ldots, Y_G \end{bmatrix}^T$, $\lambda_i = 2Nm_i \exp\left(\beta + \phi_i\right)$, $C = -\sum_{i=1}^{G} \log\left(Y_i!\right)$, and $f\left(\phi_i\right) = \frac{1}{\sqrt{2\pi}\sigma} \exp\left(-\frac{\phi_i^2}{2\sigma^2}\right)$. Note that there is no closed form for the integral in the log-likelihood function. Therefore, we use Monte Carlo integration to evaluate the log-likelihood function. Let $\phi_{ij} = \sigma\xi_{ij}$, where the $\xi_{ij}$ are independently and identically distributed random variables following a standard normal distribution. We have

$$l\left(\beta, \sigma | \boldsymbol{Y}\right) \approx l'\left(\beta, \sigma | \boldsymbol{Y}\right) = \sum_{i=1}^{G} \log\left[ \sum_{j=1}^{M} \exp\left(-\lambda_{ij}\right) \lambda_{ij}^{Y_i} \right] + C,$$

where $\lambda_{ij} = 2Nm_i \exp\left(\beta + \sigma\xi_{ij}\right)$, and $M$ is the Monte Carlo sample size which is set to be 1,000. Then, we could obtain the MLE of $\beta, \sigma$ through maximization of $l'\left(\beta, \sigma | \boldsymbol{Y}\right)$. We obtain the standard error of the MLE through inversion of the observed Fisher information matrix. However, when the DNM count is small, the Fisher information may be non-invertible and the parameter vector is not numerically identifiable. In this case, we employ group-wise jackknife using 100 randomly partitioned gene groups to obtain standard errors for parameter estimates. This approach produces consistent standard errors compared to the Fisher information approach (**Figure 5—figure supplement 9**).

The estimation procedure can be generalized to multi-trait analysis. Log-likelihood function can be expressed as follows:

$$l\left(\beta_1, \beta_2, \sigma_1, \sigma_2, \rho | Y_1, Y_2\right) = \sum_{i=1}^{G} \log \left[\int \exp\left(-\lambda_{i1} - \lambda_{i2}\right) \lambda_{i1}^{Y_{i1}} \lambda_{i2}^{Y_{i2}} * f\left(\phi_{i1}, \phi_{i2}\right) d\phi_{i1} d\phi_{i2}\right] + C,$$

where $Y_1 = \left[Y_{11}, \ldots, Y_{G1}\right]^T$, $Y_2 = \left[Y_{12}, \ldots, Y_{G2}\right]^T$, $\lambda_{i1} = 2N_1 m_i \exp\left(\beta_1 + \phi_{i1}\right)$, $\lambda_{i2} = 2N_2 m_i \exp\left(\beta_2 + \phi_{i2}\right)$, $C = -\sum_{i=1}^{G} \left[\log\left(Y_{i1}!\right) + \log\left(Y_{i2}!\right)\right]$, and $f\left(\phi_{i1}, \phi_{i2}\right) = \frac{1}{2\pi\sigma_1\sigma_2\sqrt{1-\rho^2}} \exp\left[-\frac{1}{2\sqrt{1-\rho^2}} \left(\frac{\phi_{i1}^2}{\sigma_1^2} + \frac{\phi_{i2}^2}{\sigma_2^2} - \frac{2\rho\phi_{i1}\phi_{i2}}{\sigma_1\sigma_2}\right)\right]$. We use Monte Carlo integration to evaluate the log-likelihood function. Let $\phi_{i1j} = \sigma_1 \xi_{i1j}$ and $\phi_{i2j} = \sigma_2 \left(\rho\xi_{i1j} + \sqrt{1-\rho^2}\xi_{i2j}\right)$, where the $\xi_{i1j}$ and $\xi_{i2j}$ are independently and identically distributed random variables following a standard normal distribution. We have

$$l\left(\beta_1, \beta_2, \sigma_1, \sigma_2, \rho | Y_1, Y_2\right) \approx l'\left(\beta_1, \beta_2, \sigma_1, \sigma_2, \rho | Y_1, Y_2\right) = \sum_{i=1}^{G} \log \left[\sum_{j=1}^{M} \exp\left(-\lambda_{i1j} - \lambda_{i2j}\right) \lambda_{i1j}^{Y_{i1}} \lambda_{i2j}^{Y_{i2}}\right] + C,$$

where $\lambda_{i1j} = 2N_1 m_i \exp\left(\beta_1 + \sigma_1 \xi_{i1j}\right)$ and $\lambda_{i2j} = 2N_2 m_i \exp\left[\beta_2 + \sigma_2 \left(\rho\xi_{i1j} + \sqrt{1-\rho^2}\xi_{i2j}\right)\right]$. Then, we obtain the MLE of $\beta_1, \beta_2, \sigma_1, \sigma_2, \rho$ through maximization of $l'\left(\beta_1, \beta_2, \sigma_1, \sigma_2, \rho | Y_1, Y_2\right)$. Standard error of MLE can be obtained either through inversion of the observed Fisher information matrix or group-wise jackknife if non-invertibility issue occurs.

## Computation time

Analysis of a typical pair of disorders with 18,000 genes takes about 10 min on a 2.5 GHz cluster with 1 core.

## DNM data and variant annotation

We obtained DNM data from published studies (*Supplementary file 1*-STable 1). DNM data for epileptic encephalopathies from the original release (*Allen et al., 2013*) were not in an editable format and were instead collected from denovo-db (*Turner et al., 2017*). We used ANNOVAR (*Wang et al., 2010*) to annotate all DNMs. Synonymous variants were determined based on the 'synonymous SNV' annotation in ANNOVAR; Variants with 'startloss', 'stopgain', 'stoploss', 'splicing', 'frameshift insertion', 'frameshift deletion', or 'frameshift substitution' annotations were classified as LoF; Dmis variants were defined as nonsynonymous SNVs predicted to be deleterious by MetaSVM *Dong et al., 2015*; nonsynonymous SNVs predicted to be tolerable by MetaSVM were classified as Tmis. Other DNMs which did not fall into these categories were removed from the analysis. For each variant class, we estimated the mutability of each gene using a sequence-based mutation model (*Samocha et al., 2014*) while adjusting for the sequencing coverage factor based on control trios as previously described (*Jin et al., 2017*; *Supplementary file 1*-STable 20). We included 18,454 autosomal protein-coding genes in our analysis. *TTN* was removed due to its substantially larger size.

## Description and implementation of mTADA

The method mTADA employs a Bayesian framework and estimates the proportion of shared risk genes. Specifically, mTADA assigns all genes into four groups: genes that are not relevant for either disorder, risk genes for the first disorder alone, risk genes for the second disorder alone, and risk genes shared by both disorders. The proportion of these groups are parametrized as $\pi_0, \pi_1, \pi_2, \pi_3$, respectively. In particular, parameter $\pi_3$ quantifies the extent of genetic sharing between two disorders, with a larger value indicating stronger genetic overlap (*Nguyen et al., 2020*). The 95% credible interval constructed through MCMC is used to measure the uncertainty in $\pi_3$ estimates.

The software mTADA requires the following parameters as inputs: proportion of risk genes ($\pi_1^S, \pi_2^S$), mean relative risks ($\bar{\gamma}_1^S, \bar{\gamma}_2^S$), and dispersion parameters ($\bar{\beta}_1^S, \bar{\beta}_2^S$) for both disorders. We used extTADA (*Nguyen et al., 2017*)to estimate these parameters as suggested by the mTADA paper (*Nguyen et al., 2020*). mTADA reported the estimated proportion of shared risk genes $\pi_3$ (posterior mode of $\pi_3$) and its corresponding 95% credible interval $[\text{LB}, \text{UB}]$. We considered $\pi_1^S * \pi_2^S$ as the expected proportion of shared risk genes, and there is significant genetic sharing between two disorders when

LB $> \pi_1^S * \pi_2^S$ . We quantify statistical evidence for genetic sharing by comparing the posterior distribution of $\pi_3$ with $\pi_1^S * \pi_2^S$,

$$p = 2 * \frac{\sum_{i=1}^{N_{MCMC}} I\left(\pi_3^i < \pi_1^S * \pi_2^S\right)}{N_{MCMC}},$$

where $\pi_3^i$ is the $i$-th MCMC iteration sample, $N_{MCMC}$ is the number of iterations, and $I\left(\right)$ is the indicator function. This is also equivalent to performing two-sided inference using posterior probability $P\left(\pi_3 > \pi_1^S * \pi_2^S\right)$. Number of MCMC chain was set as 2 and number of iterations was set as 10,000.

## Simulation settings

We assessed the performance of EncoreDNM under the mixed-effects Poisson model. We performed simulations for two variant classes: Tmis and LoF variants, which have the largest and the smallest median mutability values across all genes. First, we performed single-trait simulations to assess estimation precision of elevation parameter $\beta$ and dispersion parameter $\sigma$. We set the true values of $\beta$ to be −0.5,−0.25, and 0, and the true values of $\sigma$ to be 0.5, 0.75, and 1. These values were chosen based on the estimated parameters in real DNM data analyses and ensured simulation settings to be realistic. Next, we performed simulations for cross-trait analysis to assess estimation precision of enrichment correlation $\rho$, whose true values were set to be 0, 0.2, 0.4, 0.6, and 0.8. Sample size for each disorder was set to be 5000. Coverage rate was calculated as the percentage of simulations that the 95% Wald confidence interval covered the true parameter value. Each parameter setting was repeated 100 times.

We also carried out simulations to compare the performance of EncoreDNM and mTADA. False positive rate and statistical power for EncoreDNM were calculated as the proportion of simulation repeats that p-value for enrichment correlation $\rho$ was smaller than 0.05. and the proportion of simulation repeats that p-value for estimated proportion of shared risk genes $\pi_3$ was smaller than 0.05 was used for mTADA. We aggregated all variant classes together, so mutability for each gene was determined as the sum of mutabilities across four variant classes (i.e. LoF, Dmis, Tmis, and synonymous).

First, we simulated DNM data under the mixed-effects Poisson model. To see whether two methods would produce false positive findings, we performed simulations under the null hypothesis that the enrichment correlation $\rho$ is zero. We compared two methods under a range of parameter combinations of $(\beta, \sigma, N)$ for both disorders: (–0.25, 0.75, 5000) for the baseline setting, (–1, 0.75, 5000) for a setting with small $\beta$, (–0.25, 0.5, 5000) for a setting with small $\sigma$, and (–0.25, 0.75, 1000) for a setting with small sample size. We also assessed the statistical power of two methods under the alternative hypothesis. True value of enrichment correlation $\rho$ was set to be 0.05, 0.1, 0.15, and 0.2. In the power analysis, parameters $(\beta, \sigma, N)$ were fixed at (–0.25, 0.75, 5000) as in the baseline setting when both methods had well-controlled false positive rate.

To ensure a fair comparison, we also compared EncoreDNM and mTADA under a multinomial model, which is different from the data generation processes for the two approaches. For each disorder $(k = 1, 2)$, we randomly selected causal genes of proportion $\pi_k^S$ . A proportion (i.e. $\pi_3$) of causal genes overlap between two disorders. We assumed that the total DNM count to follow a Poisson distribution: $C_k \sim \text{Poisson}\left(u_k * 2N_k \sum_{i=1}^{G} m_i\right)$, where $u_k$ represents an elevation factor to represent systematic bias in the data. Let $\mathbf{Y}_k$ denote the vector of DNMs counts in the exome, $\boldsymbol{m}$ denote the vector of mutability values for all genes, and $\boldsymbol{m}_{causal, k}$ denote the vector of mutability with values set to be 0 for non-causal genes of disorder $k$. We assumed that a proportion $p_k$ of the probands could be attributed to DNMs burden in causal genes, and $1 - p_k$ of the probands obtained DNMs by chance:

$$\mathbf{Y}_k = \mathbf{Y}_{causal,k} + \mathbf{Y}_{background,k},$$

$$\mathbf{Y}_{causal,k} \sim \text{Multinomial}\left(p_k C_k, \boldsymbol{m}_{causal, k}\right),$$

$$\mathbf{Y}_{background,k} \sim \text{Multinomial}\left((1 - p_k) C_k, \boldsymbol{m}\right).$$

To check whether false positive findings could arise, we performed simulations under the null hypothesis that $\pi_3 = \pi_1^S * \pi_2^S$ across a range of parameter combinations of $(u, p, N)$ for both disorders: (0.95, 0.25, 5000) for the baseline setting, (0.75, 0.25, 5000) for a setting with small $u$ (i.e., reduced total mutation count), (0.95, 0.15, 5000) for a setting with small $p$ (fewer probands explained by

DNMs), and (0.95, 0.25, 1000) for a setting with smaller sample size. $\pi_1^S$ and $\pi_2^S$ were set as 0.1. We also assessed the statistical power of two methods under the alternative hypothesis that $\pi_3 > \pi_1^S * \pi_2^S$. In power analysis, $(u, p, N)$ were fixed at (0.95, 0.25, 5000) as in the baseline setting when false positive rate for both methods were well-calibrated.

## Comparison to the fixed-effects Poisson model

For single-trait analysis, the fixed-effects Poisson model assumes that

$$Y_i \sim \text{Poisson}\left(\lambda_i\right),$$

$$\log\left(\lambda_i\right) = \beta + \log\left(2Nm_i\right), \quad \text{for } i = 1, \ldots, G.$$

Note that the fixed-effects Poisson model is a special case of our proposed mixed-effects Poisson model when $\sigma = 0$. We compared the two models using likelihood ratio test. Under the null hypothesis that $\sigma = 0$, $2\left(l_{alt} - l_{null}\right) \sim \frac{1}{2}\chi_1^2$ asymptotically, where $l_{alt}$ and $l_{null}$ represent the log likelihood of the fitted mixed-effects and fixed-effects Poisson models respectively.

## Recurrent genes and DNMs

We used FUMA (*Watanabe et al., 2017*) to perform GO enrichment analysis for genes harboring LoF DNMs in multiple disorders. Due to potential sample overlap between the studies of developmental disorder (*Kaplanis et al., 2020*) and intellectual disability (*Lelieveld et al., 2016*), we excluded intellectual disability from the analysis of recurrent DNMs. We calculated the probability of observing two identical DNMs in two disorders using a Monte Carlo simulation method. For each disorder, we simulated exome-wide DNMs profile from a multinomial distribution, where the size was fixed at the observed DNM count and the per-base mutation probability was determined by the tri-nucleotide base context. We repeated the simulation procedure 100,000 times to evaluate the significance of recurrent DNMs. Lollipop plots for recurrent mutations were generated using MutationMapper on the cBio Cancer Genomics Portal (*Cerami et al., 2012*).

## Implementation of cross-trait LD score regression

We used cross-trait LDSC (*Bulik-Sullivan et al., 2015*) to estimate genetic correlations between disorders. LD scores were computed using European samples from the 1000 Genomes Project Phase 3 data (*Auton et al., 2015*). Only HapMap 3 SNPs were used as observations in the explanatory variable with the --merge-alleles flag. Intercepts were not constrained in the analyses.

## Estimating enrichment correlation in gene sets

Genes with a high/low probability of intolerance to LoF variants (high-pLI/low-pLI) were defined as the 4,614 genes in the upper/lower quartiles of pLI scores (*Karczewski et al., 2020*). Genes with high/low brain expression (HBE/LBE) were defined as the 4,614 genes in the upper/lower quartiles of expression in the human fetal brain (*Werling et al., 2020*). Genes with high/low heart expression (HHE/LHE) were defined as the 4,614 genes in the upper/lower quartiles of expression in the developing heart of embryonic mouse (*Zaidi et al., 2013*). Five biological pathways have been reported to be involved in congenital heart disease: chromatin remodeling, Notch signaling, cilia function, sarcomere structure and function, and RAS signaling (*Zaidi and Brueckner, 2017*). We extracted 1730 unique genes that belong to these five pathways from the gene ontology database (*Ashburner et al., 2000*) and referred to the union set as CHD-related genes. We repeated EncoreDNM enrichment correlation analysis in these gene sets. One-sided Kolmogorov-Smirnov test was used to assess the statistical difference between enrichment correlation signal strength in different gene sets.

## URLs

GWAS summary statistics data of autism spectrum disorder, schizophrenia, and Tourette disorder were downloaded on the PGC website, https://www.med.unc.edu/pgc/download-results/; Summary statistics of cognitive performance were downloaded on the SSGAC website, https://thessgac.com/; Summary statistics of epilepsy were downloaded on the epiGAD website, https://www.epigad.org/; pLI scores were downloaded from gnomAD v3.1 repository https://gnomad.broadinstitute.org/downloads; mTADA, https://github.com/hoangtn/mTADA, *Nguyen et al.,*

*2021*; denovo-db, https://denovo-db.gs.washington.edu/denovo-db/; MutationMapper on cBio-Portal, https://www.cbioportal.org/mutation_mapper; LDSC, https://github.com/bulik/ldsc; *Schorsch, 2020*.

## Code availability

EncoreDNM software is available at https://github.com/ghm17/EncoreDNM; *Guo, 2022*.

## Acknowledgements

LH acknowledges research support from the National Science Foundation of China (Grant No. 12071243) and Shanghai Municipal Science and Technology Major Project (Grant No. 2017SHZDZX01). QL acknowledges research support from the University of Wisconsin-Madison Office of the Chancellor and the Vice Chancellor for Research and Graduate Education with funding from the Wisconsin Alumni Research Foundation and the Waisman Center pilot grant program at University of Wisconsin-Madison. HZ acknowledges research support from the National Institutes of Health (Grant No. R03HD100883 and R01GM134005) and the National Science Foundation (DMS 1902903).

## Additional information

### Funding

| Funder | Grant reference number | Author |
|---|---|---|
| National Science Foundation of China | No. 12071243 | Lin Hou |
| Shanghai Municipal Science and Technology Major Project | No. 2017SHZDZX01 | Lin Hou |
| Wisconsin Alumni Research Foundation | | Qiongshi Lu |
| Waisman Center pilot grant program at University of Wisconsin-Madison | | Qiongshi Lu |
| National Institutes of Health | No. R03HD100883 and R01GM134005 | Hongyu Zhao |
| National Science Foundation | DMS 1902903 | Hongyu Zhao |

The funders had no role in study design, data collection and interpretation, or the decision to submit the work for publication.

### Author contributions

Hanmin Guo, Conceptualization, Data curation, Formal analysis, Investigation, Methodology, Resources, Software, Validation, Visualization, Writing - original draft, Writing - review and editing; Lin Hou, Qiongshi Lu, Conceptualization, Methodology, Project administration, Supervision, Writing - original draft, Writing - review and editing; Yu Shi, Formal analysis; Sheng Chih Jin, Data curation, Writing - review and editing; Xue Zeng, Boyang Li, Data curation; Richard P Lifton, Martina Brueckner, Validation; Hongyu Zhao, Methodology, Validation, Writing - review and editing

### Author ORCIDs

Hanmin Guo ⓘ http://orcid.org/0000-0001-9022-5307
Qiongshi Lu ⓘ http://orcid.org/0000-0002-4514-0969

### Decision letter and Author response

Decision letter https://doi.org/10.7554/eLife.75551.sa1
Author response https://doi.org/10.7554/eLife.75551.sa2

## Additional files

### Supplementary files
- Supplementary file 1. Supplementary Tables 1-20.
- MDAR checklist

### Data availability
The current manuscript is a computational study, so no data have been generated for this manuscript.

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
