## [Editor Report]

Lu et al. provide a powerful statistical method that measures excess sharing of de novo mutations between pairs of disorders. This method extends the concept of 'genetic correlation' to disorders caused by de-novo mutations, measuring the correlation in excess de-novo mutations in genome-wide genes for different classes of mutations. The authors apply the method to nine disorders including a developmental disorder, autism spectrum disorder, congenital heart disease, schizophrenia, and intellectual disability, finding a statistically significant overlap between 12 pairs of disorders in de novo mutations that cause a loss of gene function. This method will be of interest to researchers working on disorders caused by de-novo mutations.

---

## [Decision Letter]

**Decision letter after peer review:**

Thank you for submitting your article "Quantifying concordant genetic effects of de novo mutations on multiple disorders" for consideration by *eLife*. Your article has been reviewed by two peer reviewers, and the evaluation has been overseen by a Reviewing Editor and Molly Przeworski as the Senior Editor. The reviewers have opted to remain anonymous.

Essential revisions:

1) The manuscript needs to explain better the different approaches taken by encoreDNM and mTADA, and the corresponding strengths and weaknesses. The authors focus on power in frequentist hypothesis testing for non-zero genetic overlap between disorders, but mTADA takes a Bayesian approach. Clarification is needed here on whether the statistical basis of the comparison is fair.

2) Greater clarification of the model fitting procedure is needed: how the de-novo mutability parameter is used and the computational complexity of optimizing the likelihood function.

3) The heatmaps that present the results of the empirical analysis show only statistical significance levels. Since the authors' method's primary parameter is the correlation parameter, it would be helpful to display estimates of this parameter in the main text and to focus discussion on that parameter more and on statistical significance less.

*Reviewer #1 (Recommendations for the authors):*

Here, I have the following comments.

1. Authors proposed a mixed-effect Poisson model with overdispersion. I have a few questions regarding this model. First, it is not clear whether the de novo mutability m_i is a parameter or a given estimation in the model. Authors mentioned on line 110 that the background component is a gene-specific fixed effect but in the Methods they do not treat it as a parameter in the section of parameter estimation. Second, to treat overdispersion in Poisson model, negative binomial distribution is usually applied. Here, authors added an overdispersion variable \phi_i in the model. It will be helpful to discuss on this point. Third, a MCMC algorithm was used to solve the EncoreDNM model. How is the computational efficiency? On line 431, authors used the inversion of Fisher information matrix to obtain the covariance for parameters. Is there any non-invertibility problem here?

2. In the data analysis, authors used Figure 4c-f to suggest better fit of mixed-effect model. It is better to use a table to summarize the goodness of fit test instead of using bar plots in Supplementary Figure 3. On line 197, before reading the whole paragraph, it is not clear at the beginning whether single-trait analysis refers to mixed-effect or fixed-effect. Next, authors discovered that no significant correlation as identified between any disorders and control groups. Discussions about this should be helpful.

3. In Figure 5, panel b and c can be simplified with upper triangle for LoF and lower triangle for synoymous. Moreover, the heatmaps here only reflect the significance of correlations but no information regarding the correlation magnitude itself. It would be helpful to include this in the same figure. This point could be applied to other heatmaps in the supplement. In addition, Figure 5b shows more significant correlations for diseases with larger sample sizes. This is consistent with common sense. So including correlation magnitudes in this figure would be more informative.

Reviewer #2 (Recommendations for the authors):

1. I felt the introduction and discussion lacked a more in-depth comparison between EncoreDNM and other similar methods (specifically mTADA). In particular, what are the differences in parameters between these two methods and why do you see such an increase in false positives (Figure 3) for mTADA over EncoreDNM?

2. L66: The technical wording of this statement could be improved. The paper cited was commenting on remaining haplo-insufficient genes yet to be discovered. I also do not believe "undetected" to be the correct word here as this refers to a statistical test for enrichment rather than a clinical association. There are several genes (e.g. in Kaplanis et al. Supp Table 2) that are known to cause developmental disorders clinically but do not pass p. value thresholds suggesting a statistical enrichment in studies like DDD.

3. In Figure 1, is there a vertical line missing? On the left, there are squares representing genes, but there is also one rectangle.

4. L103-117: Could the derivation of the dispersion parameter Φ be explained in better lay-terms somewhere? I understand that it is attempting to quantify the random nature of DNM counts one expects when sequencing a subset of individuals with a given disorder, but the maths in the methods section is a bit impenetrable for somebody who is not well-versed in calculus. This is especially crucial considering the major role it plays in interpreting the various results presented, and in comparison, to mTADA. It will be difficult for some readers of a journal with a broad range of topics such as *eLife* to understand how this parameter, particularly σ, was estimated.

5. L108: Further to point 4, what are reasonable values of the parameters used to fit your model? Additionally, the authors state that β "tends to be larger… and smaller…" under various conditions, but based on Figure 3, it appears that it shifts between positive and negative?

6. L151: The word "could estimate" is loaded and represents an opinion and should be removed in favour of "estimates".

7. L184-187: The excessive use of acronyms for various disorders makes for difficult reading. While I am familiar with acronyms for developmental disorders, autism spectrum, and schizophrenia because these are my field, I constantly had to refer back to the definitions is it possible to just use the actual disorder names throughout the manuscript where possible?

8. Figure 3: I think it is relevant to put the various simulated parameters that constitute Φ (e.g. σ, ρ) into context with actual data as presented in Figure 4. i.e. do the simulated parameters used here represent reasonable assumptions of these values and are they within the expectations of mTADA? Could the range of parameters have an outsized effect on the differences between mTADA and EncoreDNM? Or does it have to do with p. value thresholding (see below). Furthermore, I would appreciate it if x-axis labels included the actual parameter setting rather than the less descriptive "small" and plots A and C had labels which described which parameters were fixed in those respective analyses.

9. L246: "i.e." should be removed. There are five genes and you listed all five.

10. L267-272: Are the authors certain the thresholding on the mTADA results is reasonable? Whilst the FDR cutoff the authors have applied may suggest "significance", the p. values for the mTADA synonymous variant analysis seem very similar across all disorder combinations. Furthermore, the general pattern of p. values for LoFs seems similar to that shown for EncoreDNM. I suspect that the correlation between p. values of EncoreDNM and mTADA will be high and one could generate similar results by adjusting significance thresholds independently for each tool.

Additionally, as mTADA is based on a Bayesian approach, shouldn't the authors threshold based on posterior probabilities rather than p. values? I am unsure if comparing the π values from the different mTADA results is a valid approach as described in the methods? How do the authors conclusions change when using thresholds based on those the authors of mTADA suggest (e.g. PP > 0.8)?

11. Figure 6: Could a visual cue/text be added to differentiate between the analyses in the upper and lower triangles?

12. What is the overall rank of genes identified between different disorders when comparing between mTADA and EncoreDNM? Could the authors plot relative p/PP values for genes identified to be significantly enriched between disorders?

13. L228-229: Do not use the terms "hints at correlation" or "correlate strongly". It either does or does not meet your p. value threshold and/or correlate.

14. L257-260: Are the recurrent LoF mutations found in developmental disorders any different than those already identified by Kaplanis et al.? If so, how do your results increase understanding of the mechanisms underlying developmental disorders? I would perhaps shift the focus of this section to identifying recurrent mutations across disorders (and perhaps cite/refer to Rees et al. in Nature Communications; PMID: 34504065).

15. L274-280: I do not feel the LD score analysis constitutes a new analysis and should be moved to the discussion. A similar result was recently published in by Abdellaoui and Verweij in Nature Human Behaviour (PMID: 33986517) that covers many of these traits.

16. Throughout: Could the term "Type I Error" be avoided? I would prefer false-discovery/false-positive be used as it is a much clearer term and immediately recognisable by the majority of readers.

17. L365-366: I do not think identifying gene-disease associations is a goal of EncoreDNM so I do not consider it a limitation of the study.

18. L584: I think the hyperlink is broken? I was able to find the repository for EncoreDNM in ghm17's github account, so this was not a major issue.

---

## [Author Response]

Essential revisions:1) The manuscript needs to explain better the different approaches taken by encoreDNM and mTADA, and the corresponding strengths and weaknesses. The authors focus on power in frequentist hypothesis testing for non-zero genetic overlap between disorders, but mTADA takes a Bayesian approach. Clarification is needed here on whether the statistical basis of the comparison is fair.2) Greater clarification of the model fitting procedure is needed: how the de-novo mutability parameter is used and the computational complexity of optimizing the likelihood function.3) The heatmaps that present the results of the empirical analysis show only statistical significance levels. Since the authors' method's primary parameter is the correlation parameter, it would be helpful to display estimates of this parameter in the main text and to focus discussion on that parameter more and on statistical significance less.

We really appreciate the thoughtful and constructive comments from the editor and both reviewers. In this revision, we have provided additional justifications for the comparison between EncoreDNM and mTADA, and clarified statistical details in the model fitting and parameter estimation procedure. We also present the enrichment correlation estimates in addition to significance levels in the updated heatmaps as suggested. The new analyses have produced highly consistent results compared to our initial submission and have strengthened the manuscript. We provide details of these analyses in the point-by-point response below.

Reviewer #1 (Recommendations for the authors):Here, I have the following comments.1. Authors proposed a mixed-effect Poisson model with overdispersion. I have a few questions regarding this model. First, it is not clear whether the de novo mutability m_i is a parameter or a given estimation in the model. Authors mentioned on line 110 that the background component is a gene-specific fixed effect but in the Methods they do not treat it as a parameter in the section of parameter estimation.

Thank you for the comment. The de novo mutability mi is given a priori rather than being a parameter to be estimated. There is extensive literature on estimating de novo mutability from genomic sequence context. In this paper, we used mutability values estimated from a tri-nucleotide-based model proposed by Samocha and colleagues^1^.

Second, to treat overdispersion in Poisson model, negative binomial distribution is usually applied. Here, authors added an overdispersion variable \phi_i in the model. It will be helpful to discuss on this point.

We thank the reviewer for pointing this out. It is true that negative binomial distribution as a Poisson-γ mixture is commonly used for handling overdispersion in count data. This is because Poisson distribution and γ distribution are a conjugate pair, and their compound model can be expressed in closed form. However, the main goal of our study is to quantify the shared genetic component of two disorders, and there is no trivial way to parametrize the bivariate extension of negative binomial distribution for this purpose. Marshall and Olkin proposed a bivariate Poisson-γ mixture (negative binomial) model with a restriction that both count variables share the same γ distribution component^2^. The mixture model has closed form solution but assumes the dispersion of two count variables to be identical and limits the unconditional correlation of two count variables to be positive.

In this paper, we adopted a Poisson-lognormal mixture framework, which is another commonly used Poisson mixture model that accommodates overdispersion (see Chapter 4.2 in Cameron, A.C. et al.^3^). Unlike the Poisson-γ mixture, this model does not have the computational convenience of having closed-form solutions. Instead, we implemented the Monte Carlo integration method to calculate the likelihood which is computationally more intensive but still obtained accurate and robust results. A major reason why we chose to use the Poisson-lognormal mixture is that it can be easily generalized to bivariate count model by replacing its univariate normal component with a bivariate normal distribution. This flexible bivariate count model was initially proposed by Munkin and Trivedi in econometrics, and has been demonstrated to accommodate both overdispersion and correlation, which suits our main goal very well^4^. In general, finding a computationally simpler approach to parametrize correlation in bivariate count data remains an interesting methodological question, but we are content with the excellent empirical performance of the Poisson-lognormal mixture model in our analyses. We have added some clarifications and discussions into the Methods-statistical model section of our revised manuscript.

Third, a MCMC algorithm was used to solve the EncoreDNM model. How is the computational efficiency?

Our Monte Carlo integration method is computationally efficient. Analysis of a typical trait pair with 18,000 genes takes about 10 minutes on a 2.5GHz cluster with 1 core. We have added these details into the Methods-computation time section in the revised manuscript.

On line 431, authors used the inversion of Fisher information matrix to obtain the covariance for parameters. Is there any non-invertibility problem here?

Thank you for raising this important point. We double checked our previous analyses and found that the non-invertibility issue indeed occurred when the DNM count is small (mostly happened when analyzing synonymous variants). In this revision, we newly implemented a group-wise jackknife method to obtain standard errors for parameter estimates when the Fisher information matrix is noninvertable. More specifically, we randomly partitioned all genes into 100 groups with equal size. Each time, we left one group out and estimated the parameters using the DNM data of the remaining genes. We then repeated the procedure 100 times and calculated the jackknife standard errors. This approach produced similar standard error estimates compared to the Fisher information approach in our analysis, as illustrated in Figure 5—figure supplement 9 in the revised manuscript. We have also incorporated the new method details into the Methods-parameter estimation section in our revised manuscript.

2. In the data analysis, authors used Figure 4c-f to suggest better fit of mixed-effect model. It is better to use a table to summarize the goodness of fit test instead of using bar plots in Supplementary Figure 3.

Thank you for your great suggestion. We have added Supplementary Table 2 in our revised manuscript to show the results of likelihood ratio tests.

On line 197, before reading the whole paragraph, it is not clear at the beginning whether single-trait analysis refers to mixed-effect or fixed-effect.

Thank you for your comment. We have added the word “under the mixed-effects Poisson model” for clarification.

Next, authors discovered that no significant correlation as identified between any disorders and control groups. Discussions about this should be helpful.

For disorders, DNMs will be enriched in risk genes and slightly depleted in non-risk genes. For control groups (these are healthy siblings recruited in a study for autism), DNMs are expected to distribute proportionally according to the de novo mutability (determined by the genomic sequence context) without showing enrichment in certain genes. Therefore, we expect the enrichment correlation, characterized by concordant enrichment of DNMs in the exome, to be near zero between disorders and the control group. Our results are consistent with our expectation. We have added some related discussions into the Results section of our revised manuscript.

3. In Figure 5, panel b and c can be simplified with upper triangle for LoF and lower triangle for synoymous. Moreover, the heatmaps here only reflect the significance of correlations but no information regarding the correlation magnitude itself. It would be helpful to include this in the same figure. This point could be applied to other heatmaps in the supplement. In addition, Figure 5b shows more significant correlations for diseases with larger sample sizes. This is consistent with common sense. So including correlation magnitudes in this figure would be more informative.

Thank you for this great suggestion. We have incorporated the enrichment correlation estimates into heatmaps (Figures 5-6). Figure 6—figure supplements 1-8 in the revised manuscript have also been updated accordingly. We also included the updated Figure 5 below for your convenience.

Reviewer #2 (Recommendations for the authors):1. I felt the introduction and discussion lacked a more in-depth comparison between EncoreDNM and other similar methods (specifically mTADA). In particular, what are the differences in parameters between these two methods and why do you see such an increase in false positives (Figure 3) for mTADA over EncoreDNM?

Thank you for the comment. To quantify shared genetic effects between two disorders, EncoreDNM assumes a mixed-effects Poisson model and estimates the correlation of deviation components across two disorders, whereas mTADA employs a Bayesian framework and estimates the proportion of shared risk genes. We have provided statistical details of EncoreDNM in the Methods section of our manuscript. Here, we briefly introduce mTADA.

The mTADA method assigns all genes into four groups: genes that are not relevant for either disorder, risk genes for the first disorder alone, risk genes for the second disorder alone, and risk genes shared by both disorders. The proportion of these groups are parametrized as π0,π1,π2,π3, respectively. In particular, parameter π3 quantifies the extent of genetic sharing between two disorders, with a larger value indicating stronger genetic overlap (for example, see Figures 4a-b in the mTADA paper^5^). The 95% credible interval constructed through MCMC is used to measure the uncertainty in π3 estimates.

Through extensive simulations and analysis of nine disorders, we demonstrated that EncoreDNM provides accurate statistical inference, but mTADA can produce false positives findings when following the author-recommended procedure, especially when the DNM count is small. One possible reason is that when there is not sufficient DNM counts, mTADA tends to overestimate π3. Nguyen et al. also reported this phenomenon in their simulation settings with small mean relative risks^5^ (see Supplementary Figure 3 in the mTADA paper). Further, although the inflation of false positive rate for mTADA may be alleviated by inducing a more stringent significance threshold, researchers will not know how to select such a threshold in practice. We will provide more details on this in our response to Comment #10 below. Related discussions have also been incorporated into the revised manuscript.

2. L66: The technical wording of this statement could be improved. The paper cited was commenting on remaining haplo-insufficient genes yet to be discovered. I also do not believe "undetected" to be the correct word here as this refers to a statistical test for enrichment rather than a clinical association. There are several genes (e.g. in Kaplanis et al. Supp Table 2) that are known to cause developmental disorders clinically but do not pass p. value thresholds suggesting a statistical enrichment in studies like DDD.

We have replaced the word “more than 1,000 genes associated with DD remain undetected” with “more than 1,000 haploinsufficient genes contributing to DD risk have not yet achieved statistical significance”.

3. In Figure 1, is there a vertical line missing? On the left, there are squares representing genes, but there is also one rectangle.

We use squares to represent different genes. The rectangle in the middle represents many other genes that are omitted due to limited space.

4. L103-117: Could the derivation of the dispersion parameter Φ be explained in better lay-terms somewhere? I understand that it is attempting to quantify the random nature of DNM counts one expects when sequencing a subset of individuals with a given disorder, but the maths in the methods section is a bit impenetrable for somebody who is not well-versed in calculus. This is especially crucial considering the major role it plays in interpreting the various results presented, and in comparison, to mTADA. It will be difficult for some readers of a journal with a broad range of topics such as eLife to understand how this parameter, particularly σ, was estimated.

We appreciate the suggestion. The main goal of EncoreDNM is to quantify correlated DNM enrichment between two disorders. The statistical framework is parametrized as follows. [Yi1Yi2]∼Poisson([λi1λi2]),log⁡([λi1λi2])=[β1β2]+log⁡([2N1mi2N2mi])+[ϕi1ϕi2],[ϕi1ϕi2]∼MVN([00],[σ12ρσ1σ2ρσ1σ2σ22]).We have described the statistical details of this framework in the Methods section of our manuscript. Briefly, in this model, DNM rate is affected by the elevation component βk, the background component log(2Nkmi), and the deviation component ϕik. The component in question, ϕik, is modeled as a random effect that follows a bivariate normal distribution. More specifically, ϕ quantifies the degree to which DNM counts look different from what we expect to see under the null (i.e., no risk genes for the disorder). A larger value of the dispersion parameter σ indicates a more substantial deviation from the null. That is, DNM counts show strong enrichment in some genes and depletion in other genes compared to the expectation based on de novo mutability. If σ has a small value, it means the DNM count data is largely consistent with the null. Parameter ρ further allows such deviation of two disorders to be correlated and is a key parameter of interest in our framework. We have added some clarifications into the Results-method overview section in the revised manuscript.

5. L108: Further to point 4, what are reasonable values of the parameters used to fit your model? Additionally, the authors state that β "tends to be larger… and smaller…" under various conditions, but based on Figure 3, it appears that it shifts between positive and negative?

In our mixed-effects Poisson regression model, there is no constraint on what value the elevation parameter β can be. A positive value of β represents over-calling DNMs while a negative value represents under-calling. The dispersion parameter σ can be any positive value. A larger σ indicates that DNM counts show a strong deviation compared to the expectation rate determined by de novo mutability. The enrichment correlation ρ quantifies the concordance of DNM enrichments between two disorders and can be any value between -1 and 1. Applying EncoreDNM to nine disorders, we found that the estimates of β were almost always negative across variant classes, which may be explained by strict quality control in DNM calling pipelines. We also note that the empirical estimates of these parameters are affected by noise in the data, especially when the sample size is small.

6. L151: The word "could estimate" is loaded and represents an opinion and should be removed in favour of "estimates".

Thank you for the suggestion. We have replaced the word "could estimate" with “estimates”.

7. L184-187: The excessive use of acronyms for various disorders makes for difficult reading. While I am familiar with acronyms for developmental disorders, autism spectrum, and schizophrenia because these are my field, I constantly had to refer back to the definitions is it possible to just use the actual disorder names throughout the manuscript where possible?

We have replaced the acronyms with the actual names of disorders throughout the revised manuscript.

8. Figure 3: I think it is relevant to put the various simulated parameters that constitute Φ (e.g. σ, ρ) into context with actual data as presented in Figure 4. i.e. do the simulated parameters used here represent reasonable assumptions of these values and are they within the expectations of mTADA? Could the range of parameters have an outsized effect on the differences between mTADA and EncoreDNM? Or does it have to do with p. value thresholding (see below). Furthermore, I would appreciate it if x-axis labels included the actual parameter setting rather than the less descriptive "small" and plots A and C had labels which described which parameters were fixed in those respective analyses.

In the real DNM data analysis using EncoreDNM, the mean(SD) of the estimated β and π3 were -0.54(0.25) and 0.99(0.36), respectively. In our simulations, β was chosen as -1, -0.5, or 0, and σ was chosen as 0.5, 0.75, or 1. Therefore the parameter values used in simulations would be reasonable and not have outsized effects on the performance of two methods. We have also added the parameter values into the x-axis labels of Figure 3.

9. L246: "i.e." should be removed. There are five genes and you listed all five.

We have removed “i.e.” from the sentence.

10. L267-272: Are the authors certain the thresholding on the mTADA results is reasonable? Whilst the FDR cutoff the authors have applied may suggest "significance", the p. values for the mTADA synonymous variant analysis seem very similar across all disorder combinations. Furthermore, the general pattern of p. values for LoFs seems similar to that shown for EncoreDNM. I suspect that the correlation between p. values of EncoreDNM and mTADA will be high and one could generate similar results by adjusting significance thresholds independently for each tool.Additionally, as mTADA is based on a Bayesian approach, shouldn't the authors threshold based on posterior probabilities rather than p. values? I am unsure if comparing the π values from the different mTADA results is a valid approach as described in the methods? How do the authors conclusions change when using thresholds based on those the authors of mTADA suggest (e.g. PP > 0.8)?

This is an important point, and we appreciate the comment. In the paper that introduced the mTADA approach, Nguyen et al. used the proportion of shared risk genes π3 to assess genetic overlaps between disorders^5^. They argued that a larger value of π3 compared to 0 indicates stronger genetic sharing (for example, see Figures 4a-b in the mTADA paper^5^), and importantly, used credible intervals for π3 to quantify the imprecision in their estimates. It is true that a PP cutoff of 0.8 was used to identify disease risk genes but this was not the approach for studying genetic overlaps in their paper.

To compare the performance of mTADA with EncoreDNM, we followed Nguyen et al. and performed posterior inference for π3. We used the same 95% credible interval approach to quantify the imprecision in π3 estimates, and compared it with π1S∗π2S which quantifies the expected proportion of shared risk genes if two disorders are genetically independent. Here, π1S and π2S are the estimated proportions of risk genes for two disorders respectively. We compared the posterior distribution of π3 with π1S∗π2S and used the following metric to quantify the statistical evidence:

p = 2∗∑i=110000I(π3i<π1S∗π2S)10000.

Here, π3i is the i-th MCMC iteration sample and I() is the indicator function. If the lower bound of the 95% credible interval for π3 exactly equals π1S∗π2S, then the corresponding p will be 0.05. This is also equivalent to using a (two-sided) 0.95 posterior probability cutoff on P(π3>π1S∗π2S) to claim statistical significance. Importantly, we once again highlight an adjustment we made in the posterior inference. Instead of comparing π3 with 0 (this is what Nguyen et al. did in the mTADA paper), comparing with the expected proportion π1S∗π2S will lead to more *conservative* inference results for π3 since π1S∗π2S is always greater than 0. Therefore, the posterior inference approach we used for mTADA is largely based on but statistically more conservative than what Nguyen et al. used for analyzing shared genetics between disorders.

Further, we investigated whether varying the significance threshold for mTADA would substantially change its performance in simulations under a mixed-effects Poisson regression model. At the significance cutoff 0.05, mTADA produced substantial proportion of false positive findings in the small β, small σ, and small N settings (Supplementary Table 14). Under a more stringent significance cutoff of 0.01, mTADA still produced a substantial inflation in false positive rates when β and σ are small. We also employed a multinomial model instead of Poisson regression to simulate DNM counts. We obtained consistent results (Supplementary Table 15).

Although it is not the analytic approach used in the mTADA paper, we also investigated an alternative strategy which makes inference based on whether two disorders share at least one risk gene with PP>0.8. This strategy produced substantial inflation in false positive rates in the baseline setting (Supplementary Tables 16-17).

The discussions above have been incorporated into the revised manuscript.

11. Figure 6: Could a visual cue/text be added to differentiate between the analyses in the upper and lower triangles?

Thank you for the constructive suggestion. We have added the text for different gene sets in Figure 6. As suggested by Comment 8 from reviewer #1, we depicted correlation estimates rather than significance levels. Figure 6—figure supplements 1-8 have also been updated accordingly.

12. What is the overall rank of genes identified between different disorders when comparing between mTADA and EncoreDNM? Could the authors plot relative p/PP values for genes identified to be significantly enriched between disorders?

EncoreDNM does not prioritize disease risk genes, but instead estimates the enrichment correlation which quantifies concordant DNM effects between two disorders. This is conceptually similar to genetic correlation which can be estimated from GWAS data. Similarly, genetic correlation quantifies the overall shared additive genetic components between two traits but does not prioritize specific SNP associations. Therefore, we are not able to compare the rank of genes between mTADA and EncoreDNM.

13. L228-229: Do not use the terms "hints at correlation" or "correlate strongly". It either does or does not meet your p. value threshold and/or correlate.

We have deleted the statement to avoid confusion.

14. L257-260: Are the recurrent LoF mutations found in developmental disorders any different than those already identified by Kaplanis et al.? If so, how do your results increase understanding of the mechanisms underlying developmental disorders? I would perhaps shift the focus of this section to identifying recurrent mutations across disorders (and perhaps cite/refer to Rees et al. in Nature Communications; PMID: 34504065).

This is a great suggestion. We identified 30 recurrent cross-disorder LoF mutations that were not recurrent in developmental disorder alone (see Supplementary Table 7 ). We have now shifted the focus of this section to cross-disorder findings. In particular, we have highlighted the gene *FBXO11* that shows recurrent LoF variants in autism and congenital hydrocephalus as an example. The Rees et al. paper has also been added as a reference in our revised manuscript.

15. L274-280: I do not feel the LD score analysis constitutes a new analysis and should be moved to the discussion. A similar result was recently published in by Abdellaoui and Verweij in Nature Human Behaviour (PMID: 33986517) that covers many of these traits.

We have moved the LD score regression analysis into the Discussion section.

16. Throughout: Could the term "Type I Error" be avoided? I would prefer false-discovery/false-positive be used as it is a much clearer term and immediately recognisable by the majority of readers.

We have replaced the word “type-I error” with “false positive rate” throughout the manuscript.

17. L365-366: I do not think identifying gene-disease associations is a goal of EncoreDNM so I do not consider it a limitation of the study.

We have removed this point from the Discussion section.

18. L584: I think the hyperlink is broken? I was able to find the repository for EncoreDNM in ghm17's github account, so this was not a major issue.

Thank you for pointing this out. The hyperlink has been fixed.

References

1. Samocha, K.E. et al. A framework for the interpretation of de novo mutation in human disease. Nature genetics 46, 944-950 (2014).

2. Marshall, A.W. & Olkin, I. Multivariate distributions generated from mixtures of convolution and product families. Lecture Notes-Monograph Series , 371-393 (1990).

3. Cameron, A.C. & Trivedi, P.K. Regression analysis of count data, (Cambridge university press, 2013).

4. Munkin, M.K. & Trivedi, P.K. Simulated maximum likelihood estimation of multivariate mixed‐Poisson regression models, with application. The Econometrics Journal 2, 29-48 (1999).

5. Nguyen, T.-H. et al. mTADA is a framework for identifying risk genes from de novo

mutations in multiple traits. Nature Communications 11, 2929 (2020).